# A Comprehensive Review of Risk Factors and Thrombophilia Evaluation in Venous Thromboembolism

**DOI:** 10.3390/jcm13020362

**Published:** 2024-01-09

**Authors:** Andrew B. Dicks, Elie Moussallem, Marcus Stanbro, Jay Walls, Sagar Gandhi, Bruce H. Gray

**Affiliations:** 1Department of Vascular Surgery, Prisma Health, University of South Carolina School of Medicine—Greenville, Greenville, SC 29601, USA; elie.moussallem@prismahealth.org (E.M.); marcus.stanbro@prismahealth.org (M.S.); sagar.gandhi@prismahealth.org (S.G.); bgray71357@gmail.com (B.H.G.); 2Department of Hematology, Prisma Health, University of South Carolina School of Medicine—Greenville, Greenville, SC 29601, USA; jay.walls@prismahealth.org

**Keywords:** deep vein thrombosis, pulmonary embolism, venous thromboembolism, risk factors, thrombophilia

## Abstract

Venous thromboembolism (VTE), which encompasses deep vein thrombosis (DVT) and pulmonary embolism (PE), is a significant cause of morbidity and mortality worldwide. There are many factors, both acquired and inherited, known to increase the risk of VTE. Most of these result in increased risk via several common mechanisms including circulatory stasis, endothelial damage, or increased hypercoagulability. Overall, a risk factor can be identified in the majority of patients with VTE; however, not all risk factors carry the same predictive value. It is important for clinicians to understand the potency of each individual risk factor when managing patients who have a VTE or are at risk of developing VTE. With this, many providers consider performing a thrombophilia evaluation to further define a patient’s risk. However, guidance on who to test and when to test is controversial and not always clear. This comprehensive review attempts to address these aspects/concerns by providing an overview of the multifaceted risk factors associated with VTE as well as examining the role of performing a thrombophilia evaluation, including the indications and timing of performing such an evaluation.

## 1. Introduction

Venous thromboembolism (VTE) is a potentially life-threatening condition characterized by the formation of blood clots in deep veins, leading to deep vein thrombosis (DVT) and the potential for pulmonary embolism (PE). VTE is a complex and multifactorial disorder influenced by a wide range of risk factors. A major theory describing the pathogenesis of VTE is Virchow’s triad which consists of the stasis of blood flow, vascular endothelial injury, and hypercoagulability [1]. With this, most identified risk factors for the development of VTE have at least one element of Virchow’s triad. 

Overall, a risk factor can be identified in the majority of patients with VTE, with the most commonly identified factors including age > 40, obesity, a personal history of VTE, and cancer [2]. These risk factors may be permanent, such as related to patients’ characteristics, or transient, such as acute clinical condition. Evidence demonstrates that the VTE risk increases proportionally to the number of predisposing risk factors [2]. Understanding the risk factors associated with VTE is important for understanding a patient’s risk of VTE development and recurrence, and thus guides providers on the best management strategies moving forward. Importantly, risk factors do not carry an equal risk of VTE development (Table 1) [2,3]. As such, physicians should consider both the strength of each individual risk factor as well as the cumulative impact of all risk factors in determining the type and duration of appropriate prophylaxis. During this evaluation, thrombophilia testing is often considered. Although these tests are readily available, it can be challenging to determine who would benefit from a thrombophilia evaluation and how the testing results will change clinical management. 

This review aims to provide an extensive exploration of these risk factors, encompassing both acquired and modifiable risk factors as well as inherited risk factors, as well as review the indications and timing for thrombophilia evaluation. 

### 1.1. Acquired and Modifiable Risk Factors

#### 1.1.1. Previous VTE

Individuals with a history of VTE are at an increased risk of recurrent thrombosis. A prospective cohort of 355 patients reported an incidence of recurrent VTE at 17.5% after two years of follow up, 24.6% after four years, and 30.3% after eight years [4]. Likewise, in a large observational study of 1231 patients with VTE, 19% of the patients reported at least one prior clinically recognized VTE event [5]. However, the risk of recurrence is highly dependent upon patient-specific factors. Patients with a history of VTE in the setting of a transient, reversible risk factor (i.e., immobilization or surgery) have a lower rate of recurrence compared to those with no known risk factors (i.e., unprovoked) or with permanent risk factors (i.e., malignancy). In the study noted above, the presence of cancer was associated with an increased risk of recurrent VTE (hazard ratio (HR) 1.72) while surgery and recent trauma or fracture were associated with a decreased risk of recurrent VTE (HR 0.36) [4]. Similarly, a prospective cohort study of 570 patients followed over 2 years noted zero recurrence of VTE in those whose first VTE occurred within six weeks of surgery compared to 19.4% recurrence in those whose first VTE had no identifiable clinical risk factors [6]. As such, while a previous VTE is a risk factor for a future VTE, the ultimate risk is highly dependent on patient-specific factors, which are further outlined below. 

#### 1.1.2. Family History of VTE

Similar to a personal history of VTE, a family history of VTE has also been identified as a risk factor for VTE development. A large national cohort study noted that having a sibling with a history of VTE incurred a relative risk (RR) of 3.08 for developing a VTE event compared to the general population [7]. It appears that the risk increases based on the number of family members with a prior VTE. In a case–control study of 505 patients, a positive family increased the risk of VTE more than 2-fold (odds ratio (OR) 2.2), with the risk increasing up to 4-fold (OR 3.9) when more than one relative has a history of VTE [8]. Interestingly, this study also noted that those with hereditary thrombophilia and a family history of VTE had a higher risk of VTE compared to those with heredity thrombophilia and no family history. Specifically, in those with a factor V Leiden mutation, a positive family history of VTE incurred a 2.9-fold higher risk compared to a negative family history [8]. These findings underscore that there are likely other inherited thrombophilias present that have yet to be discovered. 

#### 1.1.3. Immobility

Prolonged periods of immobility, such as postoperative bed rest, paralysis, hospitalization, or long-haul travel, are well-established risk factors for VTE. Immobility leads to venous stasis, particularly in the legs, which promotes thrombosis. A prior autopsy study noted that 15% of patients on bed rest for less than one week before death were found to have a venous thrombosis, with the incidence increasing to 80% for those in bed for a longer period [9]. Likewise, in a large international registry, chronically immobile elderly patients were noted to have an increased risk of recurrent VTE [10]. As immobility can be caused by numerous different factors, the risk of VTE ultimately depends on the cause and length of immobility.

The risk of VTE after an acute cerebrovascular accident (CVA) resulting in paralysis is quite high. The current rates of symptomatic VTE in patients with acute CVA ranges from 1–10%, whereas asymptomatic VTE is even higher, with a report of 11% at 10 days post CVA and 15% at 30 days post CVA [11,12,13]. Likewise, the rates of DVT within 3 months of paralytic spinal cord injury are also high, with the reported incidence of DVT being greater than 30% in those who are screened for DVT [14,15]. The risk of VTE development after spinal cord injury appears to be greatest during the first two weeks after injury, with fatal PE being rare beyond 3 months after injury [2]. Interestingly, chronic immobility in the setting of CVA or spinal cord injury does not appear to confer the same degree of risk as acute immobility. This difference is likely due to the physiologic changes that occur with chronic immobility, including leg muscle atrophy and changes in venous anatomy [2]. 

Transient immobility both during hospitalization and upon discharge to home or rehabilitation facility also represents an important risk factor for VTE. In addition to venous stasis due to immobility, acute illness can increase the risk of VTE due to increased alterations in the hypercoagulable state and damage to endothelial cells in the setting of increased inflammation. Common medical illnesses associated with VTE in hospitalized patients include infection, CVA, inflammatory bowel disease, and autoimmune diseases [16]. When compared to patients in the community, those hospitalized for any reason appear to have a 100 times greater incidence of VTE [17]. Likewise, factors associated with institutionalization, defined as current or recent hospitalization within the past three months or being a nursing home resident, independently account for over 50% of all cases of VTE in the community [18]. 

Prolonged travel, including in the car and by air, also appears to confer an increased risk of VTE. A meta-analysis of 14 studies noted that the pooled RR for VTE in travelers was as high as 2.8 [19]. Additionally, there was a dose–response relationship identified with an 18% higher risk for VTE for each 2 h increase in the duration of travel by any mode and a 26% higher risk for every 2 h of air travel. 

Lastly, prolonged sitting such as at a computer for a prolonged period also appears to confer an increased risk. In a series of patients admitted for DVT/PE, 34% reported seated immobility for a prolonged period of time (8–12 h) at work [20]. 

#### 1.1.4. Surgery

Surgical procedures have long been associated with an increased risk of VTE, as surgery can result in damage to blood vessels, activation of the coagulation cascade, and venous stasis due to immobility, both during the surgery and in the post-operative period. However, not all surgery carries the same risk of VTE, with thrombotic risk being the highest amongst orthopedic, major vascular, neurosurgery, and cancer surgery. Hip and knee arthroplasty are considered amongst the highest-risk surgeries for VTE development. Initial reports have demonstrated that the VTE incidence is as high as 30% in patients undergoing major orthopedic surgery who were not receiving thromboprophylaxis [21]. However, during more recent studies, where anticoagulation was used for VTE prophylaxis, the incidence is much lower, typically less than 5% [22,23]. The American College of Chest Physicians (ACCP) estimates the baseline perioperative, 35-day risk at 4.3% after major orthopedic surgery, with the risk highest within the first 7–14 days [24]. As such, several guidelines, including the International Consensus Meeting on VTE in 2022 (Strength of Recommendation: Strong), the American Society of Hematology in 2019 (conditional recommendation based on very low certainty), and the National Institute for Health and Care Excellence (NICE) in 2018, recommend the use of chemoprophylaxis for the prevention of VTE in this patient population [25,26,27].

In non-orthopedic surgery, open abdominal and open pelvic surgery, particularly for those associated with cancer, are also considered high risk [28,29]. Neurosurgical interventions have also reported increased rates of VTE, with a meta-analysis reporting approximately one in four patients developing VTE after neurosurgery [30,31]. Other surgeries reporting an elevated risk of VTE in the post-operative setting include coronary artery bypass, major urologic surgery, thoracic surgery, and bariatric surgery [32,33,34]. 

In contrast, laparoscopic surgery does not appear to confer the same degree of risk compared to open surgery. A retrospective study of 750,159 patients demonstrated an incidence of VTE of 0.32% within 30 days of abdominal laparoscopic surgery, with the highest incidence among patients undergoing colorectal surgery at 1.12% [35]. Similarly, another retrospective study of over 138,595 patients demonstrated that the incidence of VTE among patients undergoing laparoscopic surgery was lower compared to those undergoing open surgery (0.28% versus 0.59%, respectively) [36].

#### 1.1.5. Trauma

Trauma resulting in fracture and severe injury elevates the risk of VTE, often due to blood stasis in the setting of immobilization and via endothelial activation in the setting of injury, resulting in the activation of the clotting cascade. Like surgery, not all trauma confers the same degree of risk of thrombosis. Major trauma is associated with a significantly increased risk of VTE. A study of 716 patients with major trauma, defined as an Injury Severity Score of at least 9, who underwent screening evaluation for DVT reported a DVT incidence of 58%, with 18% occurring in the proximal veins [37,38]. Of note, these patients did not receive prophylactic anticoagulation. Interestingly, while the use of prophylactic anticoagulation does reduce the risk of VTE in patients with major trauma, the reported rates of VTE in this patient population remain high, with a reported incidence of VTE of 44% with the use of low-dose heparin and of 31% with the use of low-molecular-weight heparin [39]. Trauma resulting in fracture, particularly those involving the lower limb, is a strong VTE risk factor. The incidence differs based on the location of the fracture, with the highest risk locations including the hip (16.6%), tibial plateau (16.3%), and tibial shaft (13.3%) [1].

In contrast, minor trauma does not appear to confer the same degree of risk. In a cohort of 294 cancer-free patients with VTE admitted to hospital, the adjusted incidence rate ratio (IRR) for VTE for open wounds was 0.46 (95% CI, 0.15–1.39), for sprains 1.15 (95% CI, 0.44–3.04), and for dislocations 1.54 (95% CI, 0.37–6.48). In contrast, the adjusted IRR in the same cohort was elevated for fractures (2.45, 95% CI 1.29–4.68) and immobility (3.84, 95% CI 2.39–6.15) [40]. Likewise, a systematic review of 15 studies demonstrated an incidence of VTE of 4.8% in patients undergoing temporary lower limb immobilization due to isolated trauma [41]. 

#### 1.1.6. Cancer

Malignancy is a well-established risk factor for the development of VTE. Cancer is known to create a hypercoagulable state via the expression of hemostatic proteins on tumor cells, the release of inflammatory cytokines, and the activation of the clotting system [42]. Additionally, depending on the location and size of the tumor, the local mass effect can lead to the compression of veins with the stasis of venous flow. Amongst patients with symptomatic DVT, approximately 20% will have a known active malignancy [18,43]. The risk of cancer-associated thrombosis (CAT) varies due to several factors, including cancer site and stage, malignancy treatment, and other patient-specific factors. The risk of VTE varies broadly by cancer type. In a large registry study, the cancers associated with the highest 6-month cumulative VTE incidence were pancreatic cancer (4.4%), ovarian cancer (3.1%), Hodgkin lymphoma (2.9%), and non-Hodgkin lymphoma (2.7%); in contrast, melanoma (0.36%) and breast cancer (0.64%) were amongst the malignancies with the lowest risk [44]. Other significant risk factors for VTE development included a prior history of VTE (subdistribution HR (SHR) 7.6), distant metastasis (SHR 3.2), and the use of chemotherapy (SHR 3.4). These findings have been confirmed elsewhere with metastatic disease and the use of high-risk treatment, including surgery, radiotherapy, and chemotherapy, being associated with an increased risk of VTE [45].

The risk of VTE is highest in the first 3 months after cancer diagnosis [44,46,47]. This increased risk is likely related to cancer treatments, as several treatments, including chemotherapy, protein kinase inhibitors, antiangiogenic therapy, and immunotherapy, as well as the use of central venous catheters, have been associated with an increased risk of thrombosis [44,48]. Aside from the increased morbidity associated with VTE, CAT is reported to be the second leading cause of death after disease progression amongst patients with cancer [49]. 

Given the clear association of malignancy as a risk factor for VTE, the question often arises about screening for malignancy in a patient with VTE without other identified risk factors with the goal of the earlier detection of malignancy and thus decreasing the cancer-related mortality and improving the quality of life. Of note, the majority of cancers associated with thromboembolic events have previously been diagnosed at the time of VTE diagnosis [50]. In those without a known history of malignancy, the rate of occult cancer detection for unprovoked VTE was ~5% within 12 months of VTE diagnosis [51,52,53]. Despite this, there has been no data demonstrating improved patient-specific outcomes [53]. As such, the 2017 International Society on Thrombosis and Haemostasis recommend performing age- and gender-specific cancer screening (breast, cervical, colon, and prostate) while more intensive screening with whole-body CT or PET scan is not routinely recommended [54]. 

#### 1.1.7. Pregnancy and Postpartum

Pregnancy and the postpartum period are associated with an increased risk of VTE via several different mechanisms. Venous stasis frequently occurs in pregnancy due to the compression of the pelvic vein by the gravid uterus and due to pregnancy-associated changes in venous capacitance. Additionally, pregnancy can result in an alteration in several coagulation factors, resulting in a hypercoagulable state, as well as result in vascular injury at the time of delivery [55]. The overall incidence of VTE in pregnancy is relatively low with reports of VTE diagnosis during 1 in 1000 to 2000 pregnancies [56,57]. The incidence of DVT is reported to be three times higher than that of PE and the majority of VTE events occur in the postpartum period [56,57]. Compared to non-pregnant patients, pregnant patients have a 5-fold increased risk of VTE during pregnancy, with the risk increasing substantially to 60-fold during the first three months after delivery [58]. Additional reported risk factors associated with pregnancy-related VTE include increasing age (age > 40) and the use of assisted reproductive technology [59,60,61].

#### 1.1.8. Hormone-Based Contraception and Hormone Replacement Therapy

Estrogen-containing contraceptives and hormone replacement therapy (HRT) have been associated with an increased risk of both arterial and venous thrombosis. The mechanism is not fully understood but appears to be related to the effect that estrogen has on inducing prothrombotic and fibrinolytic changes in hemostatic factors as well as impacting the regulation of endothelial function [62]. Given their widespread use, oral contraceptives (OCPs) are one of the most important causes of thrombosis in young women. It is reported that OCPs increase the relative risk of VTE by approximately threefold [61,63,64]. The risk of VTE development with the use of OCPs appears to be highest in the first 6–12 months after the initiation of OCPs [65]. At the time of cessation of OCPs, the risk of VTE is felt to return to the level prior to OCP initiation within one to three months. Overall, the risk of VTE is considerably lower with the use of OCPs compared to the risk seen in pregnancy and the postpartum period. Additional factors that are felt to increase the risk of VTE during OCP use include smoking, obesity, polycystic ovary syndrome, older age, venous compression, and immobilization [66,67,68].

HRT is also associated with increased risk; however, this risk appears to be lower than that of OCPs, potentially due to the lower estrogen doses used in HRT compared to OCPs. Studies suggest that HRT causes an approximate twofold increase in the VTE risk [69,70,71]. Similar to OCPs, the risk of VTE development appears to be highest in the first year of HRT treatment [71]. Other risk factors associated with VTE in the setting of HRT use include older age, overweight/obesity, and factor V Leiden mutation [72]. 

#### 1.1.9. Obesity

Obesity is a recognized risk factor for VTE, likely due to its association with inflammation and the enhanced production of clotting factors. There are numerous studies demonstrating that obesity is associated with an increased risk of DVT and PE, and conversely, that underweight patients are at a reduced risk. In a study of 19,293 patients evaluating cardiovascular risk factors and venous thromboembolism, a body mass index (BMI) of greater than 40 had a sex-adjusted HR of 2.7 [73]. Likewise, a national database study demonstrated an RR of 2.5 for DVT and 2.21 for PE when comparing obese patients to non-obese patients [74]. Conversely, results from the EDITH study demonstrated underweight patients had a statistically significant reduction in risk for VTE compared with normal weight (OR 0.55) [75]. 

#### 1.1.10. Smoking

Cigarette smoking is linked to endothelial damage and inflammation and thus a heightened risk of VTE, especially in combination with other risk factors. Smoking is a well-established risk factor for atherosclerosis but has a less established link with VTE. There are several studies that have demonstrated no significant relationship between smoking and VTE [73,76]. However, others have demonstrated a link between smoking and VTE, with several demonstrating a dose-dependent link between smoking and non-smoking, with those having a higher pack year and currently smoking being at the highest risk [77,78]. 

#### 1.1.11. Age

Advancing age has been demonstrated in numerous studies to be associated with VTE, with proposed mechanisms including changes within the venous system and less effective inherent anticoagulation mechanisms. A prior study has demonstrated an exponential increase in VTE risk with age, with the annual incidence rate for DVT increasing from 17 per 100,000 persons/years for patients between the ages of 40 to 49 to 232 per 100,000 persons/year for those between the ages of 70 and 79 [79]. Similarly, it has been noted that the risk of VTE approximately doubles with each decade, starting at age 40 [2]. With this, VTEs in children and young adults are rare. When they do occur, they are usually associated with a strong predisposing risk factor, such as trauma/fracture or surgery. 

#### 1.1.12. Male Sex

Male sex has been demonstrated in several studies to be a risk factor for VTE recurrence; however, there is no reported sex differences in the risk of the first VTE event. In a meta-analysis of 2554 patients with a first VTE, the incidence of recurrence was higher in men than women, both at one year (9.5% vs. 5.3%) and at three years (11.3% vs. 7.3%) [80]. Likewise, another large meta-analysis of over 2185 demonstrated a 2.8-fold higher risk of VTE recurrence in men compared to women [81]. The mechanism behind this difference is unclear but has been reported to be due to differences in other VTE risk factors between the sexes. One prior study noted a factor V Leiden mutation as a risk factor for VTE recurrence in male patients, while the age at the first event and obesity were noted as risk factors for female patients [82]. 

#### 1.1.13. SARS-CoV-2 Disease (COVID-19)

Since the start of the COVID-19 pandemic, there have been numerous reports demonstrating an increased risk of VTE. Mechanistically, SARS-CoV-2 is felt to increase the risk of VTE via the release of proinflammatory cytokines which activate platelet aggregation, tissue factor, and the coagulation cascade, as well as via the interaction with the angiotensin converting enzyme (ACE)-2 receptor on endothelial cells, resulting in endothelial dysfunction as well as the release of vasoconstrictor angiotensin-II [83,84]. With this, numerous studies have reported increased rates of VTE in patients hospitalized with COVID-19. A large meta-analysis demonstrated that the overall prevalence of PE/DVT in hospitalized patients with COVID-19 who underwent a screening assessment for VTE was approximately 30% [85]. Moreover, a meta-analysis of twelve studies demonstrated a VTE prevalence of 31% among ICU patients, despite the use of prophylactic or therapeutic anticoagulation [86]. In contrast, the incidence of VTE in non-hospitalized patients with COVID-19 does not appear to be increased. In a large cohort of 398,000 patients, the overall incidence of VTE in non-hospitalized patients with COVID-19 was reported to be 0.1%. Likewise, in a retrospective cohort comparing COVID-19-positive patients with COVID-19-negative controls, the 30-day prevalence of VTE events was not different between the two groups (1.4% vs. 1.3%, respectively) [87]. Interestingly, it appears that the risk of VTE also differs by the strain of SARS-CoV-2 virus [88]. While there is still much left to understand about the role of COVID-19 in the VTE risk, it does appear that both the severity of COVID-19 illness and the strain of COVID-19 virus do impact the risk.

#### 1.1.14. Superficial Vein Thrombosis

As the name implies, superficial vein thrombosis (SVT) results in the thrombosis of a superficial vein. While often considered to not be as severe as DVT, studies have demonstrated that patients with SVT do have an increased risk of developing DVT. As the superficial venous system connects with the deep systems, the location of SVT does confer some risk as thrombosis near the saphenofemoral or saphenopopiteal junction is associated with an increased risk of DVT and PE development. With this, a meta-analysis of 21 studies noted that 18.1% of patients have concomitant DVT at the time of SVT diagnosis; in 11 studies, 6.9% of patients were found to have concomitant PE [89]. Longitudinally, a history of SVT also appears to carry a risk of developing DVT, with a study demonstrating that approximately one third of patients developed a DVT in four years of follow up after SVT [90]. The increased risk of developing a DVT or PE in patients with a history of SVT is likely due to shared risk factors between superficial and deep thrombosis.

#### 1.1.15. Central Vein Catheters

Intravenous catheters can lead to VTE development due to endothelial trauma and inflammation associated with catheter insertion and maintenance. The majority of SVT and DVT occurring in the upper extremities occurs in the setting of intravenous catheters [91,92]. Due to the nature of intravenous catheters, any catheter has the potential to cause venous thrombosis. In prior reports, there is a wide variation in the incidence of venous thrombosis associated with central access, ranging from 0 to 28% [93]. The risk of VTE appears to be higher with the use of peripherally inserted central catheters (PICCs) compared to a central port. Additional risk factors include active malignancy, a history of DVT, the improper positioning of catheter tip, and a subclavian venipuncture insertion site [94,95]. 

#### 1.1.16. Anatomic Risk Factors

There are several anatomic risk factors for the development of DVT. Venous compression due to anatomic variations can occur in both the upper and lower extremities, increasing the risk of VTE. In the lower extremity, May–Thurner syndrome is a common anatomic variant, resulting in the hemodynamically significant compression of the left common iliac vein between the overlying right common iliac artery and the underlying vertebral body [96]. In the upper extremity, venous thoracic outlet syndrome, also known as Paget–Schroetter syndrome, results in the compression of the subclavian vein between the first rib and a hypertrophied scalene or subclavius tendon or between the tendons themselves [97]. Compression often occurs in the setting of repetitive overhead movements, such as with weightlifting and certain sports. Both anatomic variants can lead to venous stasis and endothelial injury from repetitive compression, resulting in an increased risk of thrombosis. 

Varicose veins also appear to confer increased risk for VTE. In a large cohort of patients in Taiwan, patients with varicose veins were at an increased risk of both DVT (HR 5.30) and PE (HR 1.73) [98]. Interestingly, a population-based case–control study demonstrated that the risk of VTE associated with varicose veins appears to decrease with age: OR 4.2 at age 45, 1.9 at age 60, and 0.9 at age 75 [99].

#### 1.1.17. Other Medical Conditions

There are reports of an increased risk of VTE in patients with renal, liver, cardiovascular, and hematologic diseases. Amongst patients with renal dysfunction, chronic kidney disease, the use of hemodialysis, nephrotic syndrome, and renal transplantation have been associated with an increased risk of VTE [100,101,102,103]. As for cardiovascular disease, myocardial infarction and heart failure have been reported to be independent risk factors for VTE development [104,105]. Diabetes has also been reported as causing an increased risk of VTE, with a large meta-analysis reporting an HR of 1.35 [106]. The data on liver disease is mixed, with both increased and decreased VTE risk reported [99,107]. Myeloproliferative neoplasms, including polycythemia vera and essential thrombocythemia, are associated with both arterial and venous thrombosis [108]. Additionally, paroxysmal nocturnal hemoglobinuria (PNH) is associated with an increased risk of intrabdominal and cerebral venous thrombosis [109]. 

#### 1.1.18. Antiphospholipid Syndrome

Antiphospholipid syndrome (APS) is an acquired thrombophilia characterized by the presence of antiphospholipid antibodies, including lupus anticoagulant (LAC), beta-2 glycoprotein 1 antibodies (B2GPI), and anticardiolipin antibodies, which are directed against plasma proteins bound to anionic phospholipids [110]. These antibodies result in numerous clinical manifestations, including venous, arterial, and microcirculation thrombosis, recurrent fetal loss, and thrombocytopenia. The mechanism behind the hypercoagulability of this syndrome is multifaceted and includes inhibitions of the natural anticoagulation system, activation of procoagulant and proinflammatory effects, and activation of endothelial cells, immune cells, and the complement cascade [111,112,113,114,115]. APS may be primary or associated with systemic lupus erythematosus or other rheumatic diseases. VTE in APS typically occurs as DVT in the lower extremities; however, VTE in unusual locations, including hepatic veins, mesenteric veins, and cerebral veins are also common [116]. Amongst the antibodies, LAC is associated with this highest risk of VTE, with increasing risk with each subsequent positive antibody [117]. The risk of first VTE among asymptomatic patients with triple positive APS (positive for LAC, anticardiolipin, and anti-B2GPI) is 5.3% per year and the risk of recurrent thrombosis without anticoagulation therapy is 44% over a 10-year follow-up period [118]. 

Making a diagnosis of APS is not always straightforward. It is reported that between 2 and 5% of people in the general population have antiphospholipid antibodies without clinical sequelae [119]. Additionally, antiphospholipid antibody levels may be transiently elevated for several different reasons, including autoimmune disorders, acute infection, or chronic disease. The Sapporo criteria is useful for making the diagnosis of APS; it requires one clinical criteria and one laboratory test result that is positive on two occasions at least 12 weeks apart [120]. 

### 1.2. Inherited Thrombophilia

#### 1.2.1. Factor V Leiden Mutation

Factor V plays a role in the conversion of prothrombin to thrombin, a crucial step in the formation of blood clots. Factor V Leiden (FVL) mutation results in a point mutation in the F5 gene which encodes the factor V protein in the coagulation cascade [121]. The mutation makes factor V resistant to inactivation by activated protein C (aPC), a protein that normally helps regulate blood clotting and prevent excessive clot formation, resulting in an increased risk of VTE. Heterozygosity for FVL is the most common inherited thrombophilia in White individuals. A series of over 4000 individuals in the United States reported frequencies for FVL heterozygosity in White Americans at 5.3%, Hispanic Americans at 2.2%, Native Americans at 1.2%, African Americans at 1.2%, and Asian Americans at 0.45% [122].

Transmission is autosomal dominant and the risk of VTE differs based on patients who are heterozygous versus homozygous for the variant. Individuals with heterozygous FVL mutations infer a three- to fourfold increased risk of VTE [123,124]. In comparison, those with homozygous FVL mutations have a substantially higher risk, with reported ORs ranging from 11.5 to 79.4 [123,125]. With regards to the risk of recurrent VTE, a systematic review demonstrated that the presence of a heterozygous FVL mutation does confer only a modest increase in recurrence (OR 1.4, 95% CI 1.1–1.8) [126]. As such, most providers do not alter the long-term anticoagulation plan for a patient with heterozygous FVL. In contrast, those with a homozygous FVL mutation are typically placed on indefinite anticoagulation due to concerns for the risk of recurrent VTE. 

#### 1.2.2. Prothrombin G20210A Gene Mutation

The prothrombin G20210A gene mutation (PGM) is a gain-of-function mutation that leads to higher levels of prothrombin, and thus elevated thrombin formation, resulting in an increased risk of VTE. The G20210A point mutation in the prothrombin gene is a substitution of guanine to adenine at position 20,210 in the 3-untranslated region [127]. PGM is the second most common inherited thrombophilia after factor V Leiden, with an overall prevalence estimate of 2.0% [128]. There are geographic differences in prevalence, with prevalence being higher in individuals of European descent and very rare in individuals of Asian and African descent. 

Similar to FVL, the transmission of PGM is autosomal dominant. Individuals who are heterozygous for PGM have a three- to fourfold increased risk of VTE compared to those without the variant [127,129,130]. The data on the risk of VTE in patients who are homozygous for PGM is more limited; a small study of 36 patients with homozygous PGM reported that 33% of the patients developed VTE [131]. Interestingly, despite the increased risk associated with VTE, a systematic review of 18 articles noted that PGM heterozygosity did not confer a significant increased risk of recurrent VTE (OR 1.45, 95% CI 0.96–2.2) [132]. As such, the presence of PGM generally does not impact the decision making with regards to the duration of anticoagulation management. However, similar to homozygous FVL mutations, patients with homozygous PGM are typically recommended for indefinite anticoagulation to reduce the risk of recurrent VTE. 

#### 1.2.3. Protein C Deficiency

Protein C (PC) is an anticoagulant protein synthesized in the liver. Upon activation (aPC), the primary role of aPC is to inactive the coagulation factors Va and VIIIa, which are required for thrombin generation and factor X activation [133]. PC deficiency results in the reduced inactivation of factors Va and VIIIa, thus increasing the risk of VTE. The incidence of PC deficiency in the general population is estimated at 1 in 200 to 300 individuals [134]. In contrast, PC deficiency amongst individuals with VTE is higher, typically between 3 and 4% [135,136]. It is estimated that PC deficiency confers an approximate sevenfold increased risk of VTE [137,138]. As for VTE recurrence, a study of 130 patients with hereditary deficiencies of PC, PS, or antithrombin reported the annual incidence of recurrent VTE was 6.0% for PC deficiency [139]. The management of acute VTE in patients with inherited PC deficiency does not differ from patients without inherited thrombophilia.

#### 1.2.4. Protein S Deficiency

Protein S (PS) is a cofactor for aPC, which inactivates the procoagulant factors Va and VIIIa, reducing thrombin generation [140]. PS deficiency impairs the normal control of this mechanism, resulting in an increased risk of VTE. The prevalence of PS deficiency is difficult to interpret due to the variability in PS levels; in a cohort of 2331 adults with a personal history of VTE without a strong family history, the frequency of PS deficiency, defined as <33 units/dL, was 0.9% [141]. It is estimated the PS deficiency confers a two- to elevenfold increased risk of VTE [142]. With regards to VTE recurrence, a study of 130 patients with hereditary deficiencies of PC, PS, or antithrombin reported the annual incidence of recurrent VTE was 8.4% for PS deficiency [139]. Similar to PC deficiency, the management of acute VTE in patients with inherited PS deficiency does not differ from patients without inherited thrombophilia. 

#### 1.2.5. Antithrombin Deficiency

Antithrombin III (AT) deficiency, defined as an AT activity level consistently less than 80%, is associated with a significantly increased risk of VTE. Antithrombin is a natural anticoagulant which inhibits thrombin, factor Xa, and other serine proteases in the coagulation cascade [143]. AT deficiency can either be inherited or acquired, with acquired causes included impaired production, nephrotic losses, or accelerated consumption. Hereditary AT deficiency is relatively uncommon, with an estimated prevalence of approximately 0.2 per 1000 [144]. Compared to other thrombophilias, hereditary AT deficiency confers a much higher risk of VTE, with a prior meta-analysis demonstrating an odds ratio of VTE of 16.3 [145]. Given this increased risk, most experts recommend an indefinite course of anticoagulation to reduce the risk of recurrent thrombosis.

#### 1.2.6. Hyperhomocysteinemia

Hyperhomocysteinemia can occur by both genetic and acquired abnormality. The most common genetic defect resulting in hyperhomocysteinemia is a mutation of the enzyme methylenetetrahydrofolate reductase (MTHFR). Acquired causes include deficiencies in vitamin B6, B12, or folic acid. While older studies have reported a two- to threefold increased risk of VTE, a recent large cohort study demonstrated no increased risk of VTE in patients with elevated homocysteine concentrations [146]. Likewise, another cohort study of 478 patients reported an adjusted RR 1.6 (CI, 0.6–4.5) in patients with elevated homocysteine levels compared to those with normal levels. Additionally, the use of B vitamins to lower homocysteine levels has not been shown to reduce the recurrence of DVT or PE [147]. Consequently, measuring homocysteine levels and testing for MTHFR mutations are not recommended in patients with VTE. 

### 1.3. Thrombophilia Evaluation

Performing a thrombophilia evaluation for a patient with VTE remains a controversial issue. While these tests are readily available and typically easy to order, it can be challenging to determine who should undergo a thrombophilia evaluation and how to interpret the results. Patients with inherited thrombophilia can often be identified without testing due to several risk factors, including VTE at a young age (less than 40–50 years), a strong family history of VTE, VTE in conjunction with weak provoking factors at a young age, recurrent VTE events, and VTE in unusual sites, such as cerebral and splanchnic veins [148]. As noted in the prior sections, there are numerous acquired and inherited thrombophilias that increase the risk of VTE. Despite the associated increased risk of VTE, many studies have demonstrated that the clinical usefulness and benefits of evaluating these thrombophilias are limited, specifically as it pertains to VTE outcomes including death [148]. With this, the results of thrombophilia testing rarely impact the treatment strategy for VTE. Additionally, the significance of a positive or negative test result is often misinterpreted by clinicians. A positive test often leads to overtreatment with indefinite anticoagulation despite studies demonstrating a low risk of recurrent VTE in patients with inherited thrombophilia; in contrast, those with negative results might be missing a yet-to-be-determined thrombophilia that is not present on standard testing panels, and as such, a negative test does not always equate with low risk. With this, it is generally agreed upon that routine thrombophilia evaluation in all patients with a diagnosis of VTE is not warranted. However, there are specific patients with whom a thrombophilia evaluation might be beneficial, which are outlined below.

#### 1.3.1. Unprovoked VTE

For patients with unprovoked VTE, the risk of recurrence is known to be high, especially compared to patients with provoked VTE. The estimated rate of recurrence is approximately 10% in the first year after anticoagulation therapy is discontinued and increases to more than 50% at 10 years [149]. Interestingly, studies evaluating the risk of VTE recurrence based on thrombophilia status in patients with VTE have demonstrated no significant difference between those with and without thrombophilia. A prospective study of 474 patients without malignancy with a first VTE reported no increased risk of recurrent thrombosis in those with thrombophilia (HR 1.4; 95% CI, 0.9–2.2) [150]. Likewise, another prospective study of 570 patients with a first VTE noted that recurrence rates were not related to the presence or absence of an inherited thrombophilia (HR 1.5; 95% CI, 0.82–2.77) [6]. Lastly, the thrombophilia status of patients is unlikely to change the long-term management in those with unprovoked VTE, as guidelines recommend indefinite anticoagulation, regardless of thrombophilia status. As such, the majority of guidelines recommend against performing a thrombophilia evaluation in patients with a first unprovoked VTE event [151,152,153,154,155]. 

#### 1.3.2. Provoked VTE

Patients with VTE due to a strong, modifiable provoking risk factor, such as major surgery, trauma, hospitalization, or immobility, have a low risk of VTE recurrence, regardless of the thrombophilia status. As noted in the Surgery section above, the risk of VTE recurrence after a surgically provoked VTE is very low, with two studies demonstrating a less than 1% recurrence over a two-year period [6,156]. Given the low risk of recurrence, the presence or absence of an inherited thrombophilia is unlikely to change the anticoagulation management in these patients with recommendations for treatment for 3–6 months. As such, guidelines recommend against performing a thrombophilia evaluation in patients with a first provoked VTE event in the setting of surgery [152,153,154,155].

Aside from surgery, there are numerous other modifiable provoking risk factors for the development of VTE, including trauma, immobility, pregnancy, the use of OCPs, and hospitalization for acute medical illness. In general, while these factors are not considered to be as strongly associated with VTE risk as surgery, they do a have clear association with the development of VTE. With this, patients with provoked VTE by nonsurgical risk factors still have low rates of recurrent VTE, regardless of the thrombophilia status [157]. Similar to the recommendations for provoked VTE events in the setting of surgery, the majority of guidelines recommend against performing a thrombophilia evaluation in patients with a first provoked VTE event in the setting of a non-surgery major risk factor [152,153,154]. However, this is not agreed upon in all societal recommendations; for instance, the recently published 2023 American Society of Hematology (ASH) guidelines now recommends a thrombophilia evaluation for this with VTE provoked by a nonsurgical major transient risk factor, pregnancy or postpartum, and the use of OCP with recommendations for indefinite anticoagulation treatments in those patients with thrombophilia [155]. Of note, this is a significant change from the prior ASH recommendations published in 2013, which previously recommended against this testing. It should be noted that these are conditional recommendations based on a low level of evidence. As such, the consideration of thrombophilia testing in these patient populations should ultimately be done on an individual basis, with patients being educated on the risk/benefit of thrombophilia testing and taking into account the patient’s values and preferences.

#### 1.3.3. VTE in Unusual Sites

Cerebral and splanchnic (portal, hepatic, splenic, or mesenteric) vein thromboses are rare compared to lower extremity VTE. Thromboses in these locales have been associated with inherited thrombophilias, including FVL, PGM, and deficiencies in PC, PS, and AT [158]. The role for screening for thrombophilia in this patient population is less straightforward given the limited data and concerns for increased morbidity associated with thrombosis at these sites. As such, in those who are planning to discontinue anticoagulation after primary short-term treatment (i.e., 3–6 months), thrombophilia evaluation might be helpful to understand the risk of VTE recurrence [155]. In contrast, for those who would otherwise remain on anticoagulation indefinitely, guidelines do not recommend obtaining thrombophilia testing [155]. 

#### 1.3.4. Other Clinical Considerations

In patients with recurrent VTE, thrombophilia testing is often not necessary as it rarely changes the long-term management, given these patients have an indication for indefinite anticoagulation. However, many of these patients worry about the possibility of having an inherited thrombophilia and thus the potential risk of their offspring inheriting their thrombophilia. In this situation, a thrombophilia evaluation could be considered after properly educating the patient on the risk/benefits and implications of testing.

In young patients (age < 40) with unprovoked VTE or VTE provoked by weak risk factors, performing a thrombophilia evaluation can be considered to better understand the long-term risk of VTE recurrence. Most of these patients have an indication for an indefinite anticoagulation course; however, many young patients are not keen on being on anticoagulation for a prolonged period. With this, a thrombophilia evaluation can add further clarification to the ultimate risk of VTE recurrence, and a positive result can be used to reiterate a commitment to anticoagulation. However, it should be noted that a negative panel does not necessarily confer a lower risk of VTE recurrence and as such, long-term anticoagulation management ultimately depends on the perceived risk of VTE recurrence based on the cumulative impact of other risk factors. 

#### 1.3.5. Timing of Testing

For those undergoing thrombophilia evaluation, typical testing includes evaluation for Factor V Leiden, Prothrombin gene mutation, Protein C deficiency, Protein S deficiency, Antithrombin deficiency, and evaluation for antiphospholipid antibody syndrome. For those who are undergoing thrombophilia evaluation, the timing of the testing and the presence of anticoagulation are important considerations. Acute thrombosis can impact the levels of protein S and antithrombin, resulting in low levels which are difficult to interpret. As such, it is recommended that testing occurs outside of the acute VTE window (typically after 3 months of anticoagulation therapy). Additionally, many of the commonly used anticoagulants are known to impact the interpretability of test results, and thus it is recommended that thrombophilia testing occurs at a time when the patient is able to stop his/her anticoagulation [159]. Specifically, it is recommended to hold direct oral anticoagulants (DOACs) for 48 h and vitamin K antagonists for two weeks prior to performing thrombophilia testing [148].

## 2. Conclusions

VTE is a multifaceted condition influenced by a wide array of risk factors. There are many factors, both acquired and inherited, known to increase the risk of VTE. A risk factor can be identified in the majority of patients with VTE. However, there is heterogeneity amongst the risk factors with regards to their predictive value. As such, it is important for clinicians to understand the potency of each individual risk factor when managing patients who have a VTE or are at risk of developing VTE as this will guide counseling and management, both of the patient and their family.

## Figures and Tables

**Table 1 jcm-13-00362-t001:** Predisposing risk factors for venous thromboembolism [2,3].

Strong Risk Factors (OR < 10)
Fracture of lower limb; Hospitalization for heart failure or atrial fibrillation/flutter (within previous 3 months); Hip or knee replacement; Major trauma; Myocardial infarction (within previous 3 months); Previous VTE; Spinal cord injury.
**Moderate risk factors (OR 2–9)**
Arthroscopic knee surgery; Autoimmune diseases; Blood transfusion; Central venous lines; Intravenous catheters and leads; Chemotherapy; Congestive heart failure or respiratory failure; Erythropoiesis-stimulating agents; Hormone replacement therapy (depends on formulation); In vitro fertilization; Oral contraceptive therapy; Post-partum period; Infection (specifically pneumonia, urinary tract infection, or HIV); Inflammatory bowel disease; Cancer (highest risk in metastatic disease); Paralytic stroke; Superficial vein thrombosis; Thrombophilia.
**Weak risk factors (OR < 2)**
Bed rest > 3 days; Diabetes mellitus; Arterial hypertension; Immobility due to sitting (i.e., prolonged car or air travel); Increasing age; Laparoscope surgery (i.e., cholecystectomy); Obesity; Pregnancy; Varicose veins.

VTE—venous thromboembolism; OR—odds ratio; HIV—human immunodeficiency virus.

## Data Availability

Not applicable.

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
