# Peer review of "A Comprehensive Review of Risk Factors and Thrombophilia Evaluation in Venous Thromboembolism"

_jcm, 2024, doi:10.3390/jcm13020362_

Round 1

Reviewer 1 Report

Comments and Suggestions for Authors

The manuscript is very well written and well organized. I have a few comments and suggestions to further improve the manuscript. Please see below:

1. How did the authors come up with Table 1 and the OR stated in it? This is of importance - if these Odds ratios are stemming from meta-analysis and small prospective or retrospective studies, would delete the odds ratios as the quality of studies is not uniform. Some could be high and others low quality. Also, there could be significant heterogeneity in the study populations, making them essentially not comaparable.  

2. Line 138: "As such, guidelines recommend the use 137 of chemoprophylaxis for the prevention of VTE in this patient population" Which guidelines? Although citation is provided, it is important to state the name and year of the guideline, also would be worthy to state multiple major guidelines instead of a single reference. I see the same theme at multiple places in the manuscript, please follow the above uniformly by stating which guideline you are referring to.With the guideline recommendation, it is important to state whether it is class IA or what quality of evidence and strength of recommendation is being made. 

3. I also do not see any caveats stated about how these OR were derived, who set the standard for which OR would be considered strong and which weak. As stated above would delete the OR and put an * stating the basis on which risk factors were grouped and strong etc. 

4. For Sections 2.7 and 2.8 on Pregnancy and Post-partum and Hormone replacement Therapy: These important papers reviewing VTE in women, and especially in the pregnant population should be discussed:

"Thachil R, Nagraj S, Kharawala A, Sokol SI. Pulmonary Embolism in Women: A Systematic Review of the Current Literature. Journal of Cardiovascular Development and Disease. 2022 Jul 25;9(8):234."
"Kearsley R, Stocks G. Venous thromboembolism in pregnancy-diagnosis, management, and treatment. BJA Educ. 2021 Mar;21(3):117-123. doi: 10.1016/j.bjae.2020.10.003. Epub 2021 Jan 6. PMID: 33664981; PMCID: PMC7892350."

5. "Additionally, many of the com- 575 monly used anticoagulants are known to impact the interpretability of testing results, and 576 thus it is recommended that thrombophilia testing occur at a time when the patient is able 577 to stop his/her anticoagulation" - this is a bit vague. What about those on life-long AC? If it is reasonable to stop AC for 48-72 hours (based on renal function) after the acute phase of VTE has passed and then do the testing, please state that vs. what is usually recommended.

6. Is there any disagreement between the guidelines in the context of thrombophilia testing, VTE risk factors? If there is discrepancy then I suggest making a table stating it all at one place with the level of evidence, so the reader can get the gist. 

Reviewer 2 Report

Comments and Suggestions for Authors

In the review the authors summarize findings regarding detect ion and management of VTE. Although the review seems to be well-structurized and gives comprehensive understanging of VTE therapy, I wold like to make several comments to discuss.

1. The authors should give thorough and consice algoruthm of the detection of VTI risk based on their arguments of established reasons and thrombophilia. Genetic and epigenetic causes should be also discussed in connection with VTE step-by-step decision-making.

2. Please, concentrate on the risk of VTE and provide separate management among the individuals with various risk values.

3. The section Conclusion requires to be more practically useful than it is. Please, chack and correct it.
